# Identification of Genes Related to Squab Muscle Growth and Lipid Metabolism from Transcriptome Profiles of Breast Muscle and Liver in Domestic Pigeon (*Columba livia*)

**DOI:** 10.3390/ani12091061

**Published:** 2022-04-20

**Authors:** Zhaozheng Yin, Wei Zhou, Haiguang Mao, Xinyang Dong, Xuan Huang, Haiyang Zhang, Honghua Liu

**Affiliations:** College of Animal Science, Zijingang Campus, Zhejiang University, Hangzhou 310058, China; yzhzh@zju.edu.cn (Z.Y.); 21917002@zju.edu.cn (W.Z.); maohaiguang@163.com (H.M.); sophiedxy@zju.edu.cn (X.D.); 22017085@zju.edu.cn (X.H.); 22017062@zju.edu.cn (H.Z.)

**Keywords:** pigeon, muscle growth, lipid metabolism, transcriptome, gene

## Abstract

**Simple Summary:**

Domestic pigeon is an important small poultry species raised for high-quality meat production. However, the relevant gene associated with meat growth and lipid metabolism during the period from dehulling to marketing are not known. Therefore, we aim to identify genes related to squab muscle growth and lip metabolism from transcriptome profiles of breast muscle and liver in domestic pigeon. In this study, we totally found that 4465 differentially expressed genes (DEGs) identified in the breast muscle and liver libraries, which include 2585 genes that were up-regulated and 2122 genes that were down-regulated. Most genes are involved in cell proliferation and differentiation, lipid metabolism and energy metabolism according to Gene Ontology (GO) and Kyoto Encyclopedia of Genes and Genomes (KEGG) enrichment analysis of DEGs. We also detected 16 DEGs to verify data from RNA-Seq and qPCR, which were consistent in quantitatively estimating the transcription levels of the tested transcripts by qPCR analysis. The results of this study will lay the foundation for understanding the mechanisms of muscle growth and lipid metabolism in domestic pigeons.

**Abstract:**

The improvements in muscle growth rate and meat quality are the major breeding aims in pigeon industry. Liver and muscle are recognized as important sites for fatty acid metabolism; understanding the role of specific transcripts in the breast muscle and liver might lead to the elucidation of interrelated biological processes. In this study, RNA-Sequencing (RNA-Seq) was applied to compare the transcriptomes of breast muscle and liver tissues among pigeons at five developmental periods (0, 1, 2, 3, 4 weeks post-hatching) to identify candidate genes related to muscle growth and lipid metabolism. There were 3142 differentially expressed genes (DEGs) identified in the breast muscle libraries; 1794 genes were up-regulated while 1531 genes were down-regulated. A total of 1323 DEGs were acquired from the liver libraries, with 791 up-regulated genes and 591 down-regulated genes. By pathway enrichment analysis, a set of significantly enriched pathways were identified for the DEGs, which are potentially involved in cell proliferation and differentiation, lipid metabolism and energy metabolism in pigeon breast muscle and liver. Our results are consistent with previous partial reports from domestic animals and poultry and provide some unidentified genes involved in muscle growth and lipid metabolism. The reliability of the sequencing data was verified through qPCR analysis of 16 genes from eight comparison groups (two genes per group). The findings from this study could contribute to future investigations of muscle growth and lipid metabolism mechanisms and establish molecular approaches to improve muscle growth rate and meat quality in domestic pigeon breeding.

## 1. Introduction

Meat-type pigeon, especially the squab (young domestic pigeon of approximately 4 weeks old), has been consumed by many nations for centuries for the high nutritious value [1,2]. The pigeon breeding programs need to focus on the increase of growth rate and meat quality, such as tenderness, intramuscular fat (IMF) and fatty acid compositions, to meet the growing demand of consumers. It is important to note that traits determining meat quality are difficult to improve by the traditional selection because the heritability is moderate to low and the measurement of the meat quality traits are difficult, expensive and only possible after slaughter [3]. Therefore, the identification and characterization of the interacting genes which regulate muscle growth and meat quality are helpful for pigeon production, which might lay the foundation for the program of marker-assisted selection (MAS).

The development of RNA sequencing (RNA-Seq) provided a new platform for researchers to efficiently identify the novel and low-abundance transcripts via transcriptome profiling [4,5]. The extensive application of this digital gene expression technology in different tissues or breeds provided a huge amount of information about the mRNA transcriptome [6,7,8,9,10].

Muscle and liver are important organs involved in the development of intramuscular adipose tissue [11]. Depending on the species, the liver is more or less the center of fatty acid synthesis and lipid recycling through lipoprotein synthesis [12]. In addition, previous studies have clearly revealed that the total number of muscle fibers remained unchanged and the muscle growth mainly depended on the hypertrophy of muscle fibers after birth [13,14,15], which was also accompanied by a dramatic increase in IMF [16]. Hence, the gene discovery of muscle growth and development during the early period after birth could provide vital information for the molecular mechanisms that determine meat quality. To our best knowledge, few studies have investigated the expression difference of genes in meat-type pigeon during post-hatching four weeks. In this study, we used the next-generation sequencing to generate transcriptome profiles of muscle and liver tissue to identify the functional genes and the regulation networks that control muscle growth and lipid metabolism in domestic pigeon. Results obtained here could contribute to understanding the background of molecular genetic mechanisms involved in muscle growth and lipid metabolism in pigeons.

## 2. Materials and Methods

All the animal procedures were carried out in strict accordance with the guidelines from the Care and Use Committee at the College of Animal Science of Zhejiang University (Approval Number: ZJU20190149).

### 2.1. Animals and Sample Collection

The pigeons used in this study were obtained from Jiangyin Weitekai pigeon industry Co. Ltd (Wuxi, China). A total of 25 female squabs (five birds per week) were randomly selected at 0 w, 1 w, 2 w, 3 w and 4 w post-hatching. The birds were reared by parental pigeons until 4 weeks of age, and their parents were housed one pair (male–female paired) per cage under the same managerial conditions. Each pair of parental pigeons fed two squabs. Feed and water were provided ad libitum. Breast muscle and liver tissues were collected from five randomly selected pigeons at different stages and immediately flash frozen in liquid nitrogen until further use.

### 2.2. RNA Extraction

Total RNA was isolated from the breast muscle and liver tissues using the TRIzol reagent (Invitrogen, Waltham, MA, USA) according to the manufacturer’s instructions. The degradation and contamination of the RNA were monitored on 1% agarose gels. The RNA integrity was assessed using the RNA Nano 6000 Assay Kit of the Bioanalyzer 2100 system (Agilent Technologies, Santa Clara, CA, USA). The concentration of the total RNA was measured using a Qubit^®^ RNA Assay Kit in Qubit^®^ 2.0 Fluorometer (Life Technologies, Carlsbad, CA, USA) and the purity was checked using the NanoPhotometer^®^ spectrophotometer (IMPLEN, Westlake Village, CA, USA).

### 2.3. Library Construction and Sequencing

A total of 50 libraries (*n* = 5) were constructed and sequenced from 50 samples including 25 breast muscle tissues (5 birds × 5 time periods) and 25 liver tissues (five birds × 5 time periods) by the Novogene Bioinformatics Technology Co., Ltd. (Beijing, China). For the RNA sample preparations, 1 µg RNA per sample was used as input material. Sequencing libraries were generated using an NEBNext^®^ Ultra™ RNA LibraryPrep Kit for Illumina^®^ (NEB, Ipswich, MA, USA) following the manufacturer’s recommendations, and index codes were added to attribute sequences to each sample. Subsequently, the final products were purified (AMPure XP system, Beckman Coulter, Brea, CA, USA) and library quality was assessed on the Agilent Bioanalyzer 2100 system (Agilent Technologies, Inc., Santa Clara, CA, USA). The clustering of the index-coded samples was performed on a cBot Cluster Generation System using TruSeq PE Cluster Kit v3-cBot-HS (Illumia, San Diego, CA, USA) according to the manufacturer’s instructions. After cluster generation, the library preparations were sequenced on the Illumina sequencing platform (HiSeq^TM^ 2000, Illumia, San Diego, CA, USA) and 125 bp/150 bp paired-end reads were generated.

### 2.4. Data Filtering and Mapping of Reads

Clean data were obtained by removing reads containing an adapter, reads containing ploy-N and low quality reads from raw data. At the same time, Q20, Q30 and GC content of the clean data were calculated [17]. All clean sequencing reads were mapped to the reference genome of pigeon from NCBI using Hisat2 v2.0.4 software [18]. FeatureCounts v1.5.0-p3 was used to count the reads numbers mapped to each gene [19]. Then, the expected number of fragments per kilobase of transcript sequence per millions base pairs sequenced (FPKM) of each gene was calculated based on the length of the gene and the reads count mapped to this gene [20].

### 2.5. Differential Gene Analysis and Functional Annotation

String Tie was used to perform the expression level for mRNAs by calculating FPKM. The differentially expressed genes (DEGs) mRNAs were selected with log2 (fold change) > 1 or log2 (fold change) ≤ 1 and with statistical significance (*p* value < 0.05) by R package–Ballgown [21]. FPKM analyzed by HTSeq v0.9.1 was used as the value of the normalized gene expression [22]. The false discovery rate was controlled by adjusting the *p*-values using the Benjamini–Hochberg method [23]. Gene Ontology (GO) enrichment analysis of DEGs was implemented by the GOseq R package [24]. The KOBAS software was used to test the statistical enrichment of differential expression genes in the Kyoto Encyclopedia of Genes and Genomes (KEGG) pathways [25]. KEGG pathways and GO terms, including biological process (BP), cellular component (CC) and molecular function (MF) with a corrected *p*-value of less than 0.05 were considered to be significantly enriched by DEGs. DESeq provides statistical routines for determining differential expression in digital gene expression data using a model based on the negative binomial distribution. The resulting *p*-values were adjusted using Benjamini and Hochberg’s approach for controlling the false discovery rate.

### 2.6. Gene Expression Analysis by RT-qPCR

Sixteen DEGs (2 genes per comparison group) were selected randomly for the purpose of validating the results of the sequencing, which was tested by real-time fluorescent quantitative PCR (RT-qPCR) on the StepOnePlus Real-time PCR System (Applied Biosystems, Waltham, MA, USA) using SYBR Premix PCR kit (TIANGEN, Beijing, China). The 10 μL PCR reaction mixture included 1 μL of the cDNA, 5 μL 2 × SuperReal Color PreMix, 1 μL 50 × ROX Reference Dye, 2.4 μL RNase-free ddH2O, 0.3 μL forward primer and 0.3 μL reverse primer. Reactions were incubated in a 96-well optical plate (Applied Biosystems, Waltham, MA, USA) at 95 °C for 15 min, 40 cycles of 95 °C for 10 s and 60 °C for 30 s. Primers (Appendix A) were designed using Premier 5.0 (Premier Biosoft International, Palo Alto, CA, USA) according to the pigeon mRNA sequences in GenBank. Three independent biological replicates were used to analyze all mRNA expression. The expression levels of the mRNA were normalized to the mRNA expression of pigeon β-actin and were calculated using the 2^−ΔΔCt^ method [26]. The relative expression data were checked for normality by the Kolmogorov–Smirnov test in statistical software SPSS v20.0 (IBM, Chicago, IL, USA), and logarithmic transformation (log2) was used to correct the non-normal distribution. Then, one-way ANOVA was used to analyze the relative expression of the gene among groups. Multiple comparison results of the groups were corrected by Bonferroni correction.

## 3. Results

### 3.1. Sequencing Data and Read Mapping

In this study, 48.04 M to 81.97 M raw reads were generated for each sample with the average base quality above Q30 for 93% of bases. Mean GC content (guanine and cytosine ratio) of the liver and muscle transcriptome were found to be 53.61% and 53.91%, respectively. Additionally, 72.55% to 78.38% of the clean reads were aligned with the pigeon reference genome (Appendix A and Table 1). If the FPKM values of all the detected genes were divided into five intervals, less than 1, 1 to 3, 3 to 15, 15 to 60, and more than 60, the distribution of the FPKM of the genes was shown to be similar among the five libraries (Appendix A). When the filter criteria of FPKM > 1 and *p*-value < 0.05 was applied, 3142 DEGs were acquired from the breast muscle libraries. Of these DEGs, 1794 genes were up-regulated and 1531 genes were down-regulated. In the liver libraries, a total of 1323 DEGs were identified, with a total of 791 up-regulated and 591 down-regulated genes during different development stages (Figure 1 and Figure 2).

### 3.2. Cluster Analysis of DEGs

Gene expression patterns at the scale of the transcriptome were measured by systematic cluster analysis to explore the similarities and compare the relationship between the different libraries in breast muscle tissue and liver tissue. The results of hierarchical clustering indicated that the gene expression similarity could be classified into distinct groups. BM1 and BM2 were clustered together, and BM3 and BM4 were of the same class; L3 and L4 had similar expression patterns while L0, L1 and L2 were clustered into respective class (Figure 3).

### 3.3. GO Analysis for DEGs

To further investigate the functional annotation of the breast muscle and liver transcriptomes, the DEGs were summarized into three categories of biological process, cellular component and molecular function at the significant level of the corrected *p*-value < 0.05 (Appendix A). The top 30 GO terms of DEGs of each comparison library in the breast muscle and liver were presented in Figure 4. GO analysis showed that DEGs were associated with ATP metabolic process, myosin complex, mitochondrial inner membrane, proteinaceous extracellular matrix, microtubule binding, mitotic M phase and hydrogen ion transmembrane transporter activity for breast muscle. In the liver, DEGs were involved in GO terms such as ATP binding, hydrolase activity, acting on acid anhydrides, cell cycle process and microtubule.

### 3.4. Pathway Enrichment Analysis of DEGs

To identify the biological pathways involved in muscle growth and lipid metabolism, we mapped the DEGs to terms in the KEGG database. As shown in Figure 5, the 20 most enriched pathways were identified separately from eight different comparison groups in breast muscle and liver. In breast muscle tissue, DEGs showed significant enrichment in process relevant to metabolic pathways, oxidative phosphorylation, citrate cycle (TCA cycle), glycolysis/gluconeogenesis, biosynthesis of amino acids, insulin signaling pathway and ECM–receptor interaction. While cell cycle, fatty acid degradation, p53 signaling pathway, biosynthesis of unsaturated fatty acids etc. were found to be more associated with the DEGs in the liver. These differences provided some cues to survey the spatial and temporal expression of genes related to muscle growth and lipid metabolism.

### 3.5. Validation of DEGs

We selected 16 DEGs (myosin heavy chain 7 (*MYH7*), fat storage inducing transmembrane protein 2 (*FITM2*), insulin receptor substrate 2b (*IRS2B*), CD1d molecule (*CD1D*), calmodulin dependent protein kinase ID (*CAMK1D*), collagen type IV alpha 6 chain (*COL4A6*), signal peptide (*SCUB1*), fatty acid binding protein 3 (*FABP3*), minichromosome maintenance complex component 3 (*MCM3*), cytochrome P450 family 3 subfamily A member 28 (*CYP3A28*), colipase (*CLPS*), acyl-CoA oxidase 2 (*ACOX2*), glutamine-fructose-6-phosphate transaminase 2 (*GFPT2*), cyclin B2 (*CCNB2*), C-X3-C motif chemokine ligand 1 (*CX3C*) and ArfGAP with coiled-coil, ankyrin repeat and PH domains 3 (*ACAP3)*) randomly from 8 comparison groups (two genes per group) to validate the accuracy of the RNA-Seq using qPCR, including the up- and down-regulated genes. The qPCR analysis confirmed the direction of changes in the expression level of DEGs detected by RNA-Seq (Figure 6), which demonstrated that data from RNA-Seq and qPCR were consistent in quantitatively estimating the transcription levels of the tested transcripts. Moreover, we detected a temporal expression profile of six DEGs in breast muscle and liver (Figure 7). In breast muscle, the *FITM2* gene was highly expressed at 2 weeks, the *CAML1D* gene at 0 and 4 weeks, and the *COL4A6* gene at 0 weeks. In the liver, the *MCM3* and *CCNB2* genes were highly expressed at 0 weeks and the *ACAP3* gene was highly expressed at 1 and 3 weeks.

## 4. Discussion

Since parents only raise growing young birds to a mature weight of 4 weeks, 0–4 weeks is a critical growth period for squabs [1]. However, there are relatively few studies providing the genetic background of muscle development and lipid metabolism in squabs from a tissue transcriptome perspective. Interestingly, we found that DEGs had different levels of enrichment between the distinct comparison libraries of a week of age. Many DEGs were detected in breast muscle and liver in the first three weeks post-hatching and identified here were involved in these signaling pathways, including the MCM family (*MCM2*, *MCM3*, *MCM5*, *MCM6*), CCNB family (*CCNB2*, *CCNB3*), *ORC1* and *CDK1*. In addition, cluster analysis showed that the gene expression patterns at the third and fourth week were similar in both the breast muscle and liver tissue of pigeons, which indicated that pigeons had a more complex regulatory mechanism of growth and development at the early stage. The function of these transcripts should be further validated by additional biological analysis and experimental evidence.

In the first four weeks post-hatching, many DEGs (enolase 1 (*ENO1*), ATP synthase (*ATP5B*), putative NADH dehydrogenase [ubiquinone] iron-sulfur protein (*NDUF*), Sorbitol dehydrogenase (*SDH*), regulatory cox family protein (*COX*), dihydrolipoamide S-acetyltransferase (*DLAT*), acyl-CoA synthetase (*ACSS*), dihydrolipoamide dehydrogenase (*DLD*), pyruvate dehydrogenase E1 subunit beta (*PDHB*), fructose-bisphosphatase 2 (*FBP2*), protein phosphatase 1 regulatory subunit 3B (*PPP1R3B*)) in breast muscle were enriched in several signaling pathways, such as the oxidative phosphorylation, citrate cycle (TCA cycle), glycolysis/gluconeogenesis, pyruvate metabolism and insulin signaling pathways. More and more studies have shown that genes related to energy metabolism play an important role in the process of muscle cell growth and myofibrillar proteins transformation [27,28]. Moreover, it has been shown that, in response to energy limitation, adaptive reductions in whole-body energy expenditure are linked to structural and functional changes in mitochondria in the liver and skeletal muscle [29]. In addition, there was a close relationship between the ATP gene and energy metabolism of animals, skeletal muscle formation and IMF content. As a member of the ATP synthase family, *ATP5B* was not only involved in lipid metabolism as high-density lipoprotein receptors in the liver cells, but also participated in the regulation of skeletons’ muscle formation in combination angiostatin [30]. Moreover, *PPP1R3B* could stimulate glycogen accumulation and decrease muscle glycogen phosphorylase (GP) phosphatase activity. Additionally, *PPP1R3B* gene was highly expressed in human skeletal muscle and liver which suggested that *PPP1R3B* could be a functional candidate gene for muscle development [31]. Muscle with high glycogen and lactate content showed rapid post-mortem glycolysis, paler surface color, high drip loss and higher extents of protein denaturation [32]. Presumably, *PPP1R3B* might be associated with meat quality and further investigation with a large study was needed to confirm it.

It was well known that lipid metabolism was critically important in determining fat deposition. Several pathways relating to lipid metabolism were identified in our study, including steroid hormone biosynthesis, steroid biosynthesis, fatty acid degradation, fatty acid metabolism, linoleic acid metabolism etc., which formed a network to influence IMF content in the breast muscle post-hatching. DEGs involved in these pathways might be suggested as promising functional and positional candidate genes for fat deposition, such as *ACOX2*, *ACOX3, CYP3A28*, *CYP7A1*, hydroxysteroid 17-beta dehydrogenase 7 (*Hsd17b7*), pancreatic lipase related protein 1 (*PNLIPRP1*), aldehyde dehydrogenase 3 family member A2 (*ALDH3A2*), NAD(P) dependent steroid dehydrogenase-like (*NSDHL*), alanine--glyoxylate and serine--pyruvate aminotransferase (*AGXT*), phospholipase A2 group IB (*PLA2G1B*). Pathway analysis showed that *ACOX3* was involved in fatty acid degradation and glycometabolism. As an important oxidase, *ACOX3* participated in the fatty acid β-oxidation of peroxisome [33]. The oxygenolysis of fatty acids not only supplies energy to the organism, but also provides materials for the synthesis of proteins, carbohydrate and lipids. In mammals, long-chain, very long chain and branched chain fatty acids could be oxidized and decomposed in the peroxisome. Dysfunction of peroxidosome could lead to excessive deposition of fatty acids and metabolic diseases [34]. Therefore, the ACOX3 gene might be one of the important genes affecting the oxidation of fatty acids in pigeons. In addition, Cholesterol 7a-hydroxylase (*CYP7A1*) was a member of the cytochrome P450 (CYP) superfamily, which constituted a multigene membrane-bound enzyme system to catalyze the oxidation of relevant endogenous compounds, such as steroids and bile acids [35]. *CYP7A1* was a rate-limiting enzyme in bile acid synthesis. It was likely to play a crucial role in post-hatching development and survival. In humans, there was a significant association between the *PLA2G1B* gene polymorphisms and adiposity. This gene encoded the Group 1B phospholipase A2 protein, which was the primary lipase that hydrolyzes lipids, converting triglyceride substrates to monoglycerides and free fatty acids [36]. All the genes mentioned here should be studied in detail in the future.

Consistent with the previous study [37], several DEGs in breast muscle and liver were enriched in the ECM-receptor interaction signaling pathway, such as *COL4A5*, *COL4A6*, *COL6A6*, *COL6A3*, *THBS2*, *THBS3*, *AGRN*. As structural components of the extracellular matrix (ECM), collagen (COLA) turnover has been observed to be accelerated during the periods of rapid growth. Variation in the solubility of collagen also has been associated with meat tenderness [38]. It could be inferred that *COL4A5*, *COL4A6*, *COL6A6* and *COL6A3* might be related to meat quality. In addition, some well-known pathways reported in other studies did not reach significant enrichment levels in our data, such as the PPAR signaling pathway, Jak-STAT signaling pathway and MAPK signaling pathway. Cui et al. suggested that the deposition of intramuscular fat in chicken was associated with the PPAR pathway [39]. Both the Jak-STAT signaling pathway and the MAPK signaling pathway played a vital role in muscle fibers specialization and muscle mass maintenance [40,41]. Two DEGs, the signal transducer and activator of transcription (*STAT*) and adiponectin (*ADIPO*), were selected from the three pathways to be plausible candidate genes. Previous studies have demonstrated the role of STIAT2 in muscle production and lipid metabolism [42,43]. AdipoR2 has been shown to inhibit lipogenesis, as well as activate PPAR-α and fatty acid oxidation genes [44,45].

In summary, the findings from our study provided a new insight into transcriptional profiles of muscle development and lipid metabolism at five stages of domestic pigeon using a deep sequencing method. Compared with the first three weeks, changes in muscle growth and lipid metabolism of pigeons were no longer intense by the fourth week post-hatching. Many DEGs related to cell proliferation and differentiation, lipid metabolism and energy metabolism indicated that a complicated gene network might be involved in muscle growth and fat deposition. The enrichment of GO terms and KEGG pathways in our datasets would provide valuable information for understanding muscle development mechanisms and for finding molecular approaches to improve meat quality traits in pigeon breeding. The function of these DEGs provided by our results should be further validated by additional biological analysis and experimental evidence.

## 5. Conclusions

In this study, we identified 4465 differentially expressed genes through transcriptome sequencing analysis in the breast muscle and liver of domestic pigeon at different growth stages. We predicted 16 genes involved in muscle growth and lipid metabolism—MYH7, FITM2, IRS2B, CD1D, CAMK1D, COL4A6, SCUB1, FABP3, MCM3, CYP3A28, CLPS, ACOX2, GFPT2, CCNB2, CX3C and ACAP3—through metabolic pathways, cell cycles and neuroactive ligand-receptor interaction. The results of this study will help to investigate the regulatory network and molecular mechanisms of muscle growth and lipid metabolism in domestic pigeons, and will provide valuable information for establishing molecular approaches to improving muscle growth rate and meat quality in domestic pigeon breeding.

## Figures and Tables

**Figure 1 animals-12-01061-f001:**
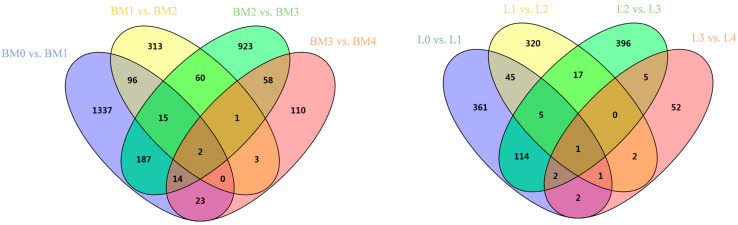
Venn diagrams of the number of DEGs in each period comparison (0, 1, 2, 3 and 4 weeks post-hatching) of breast muscle and liver. BM: breast muscle; L: liver.

**Figure 2 animals-12-01061-f002:**
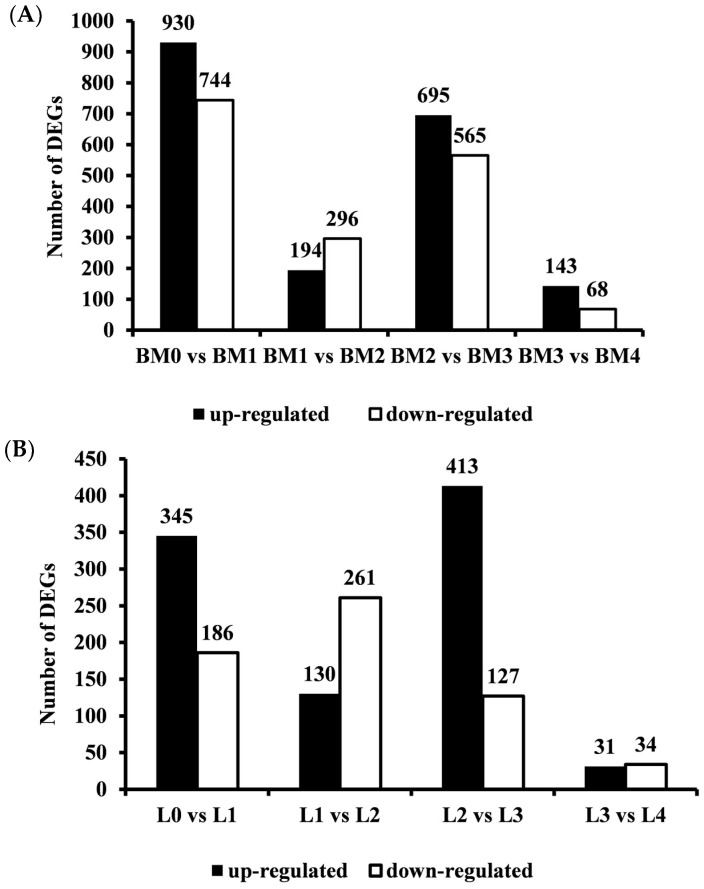
Number of DEGs between the comparison libraries. BM: breast muscle; L: liver. 0, 1, 2, 3 and 4 weeks post-hatching. (**A**): breast muscle; (**B**): liver.

**Figure 3 animals-12-01061-f003:**
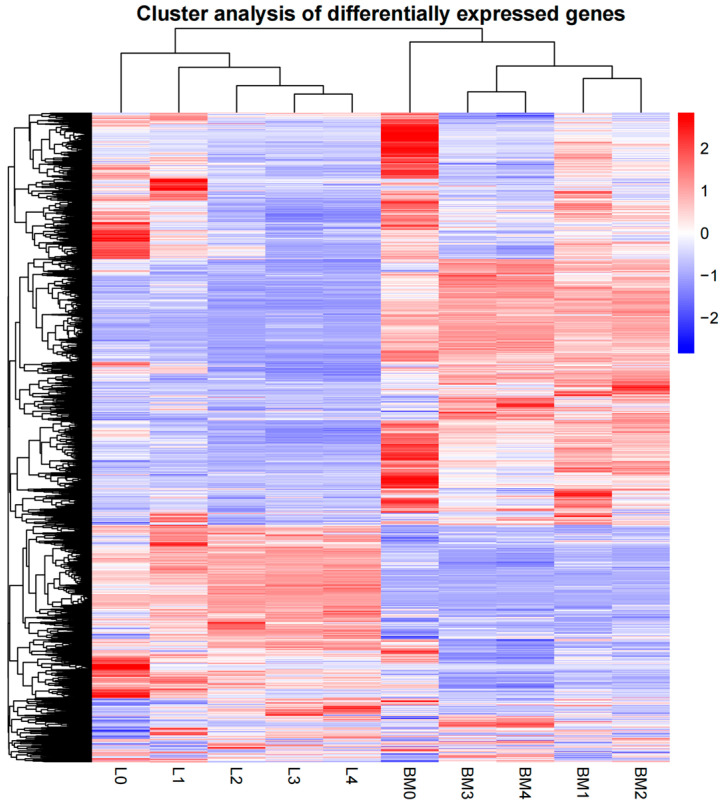
Hierarchical clustering analysis for all the DEGs in breast muscle and liver tissues. 0, 1, 2, 3 and 4 weeks post-hatching. BM: breast muscle; L: liver.

**Figure 4 animals-12-01061-f004:**
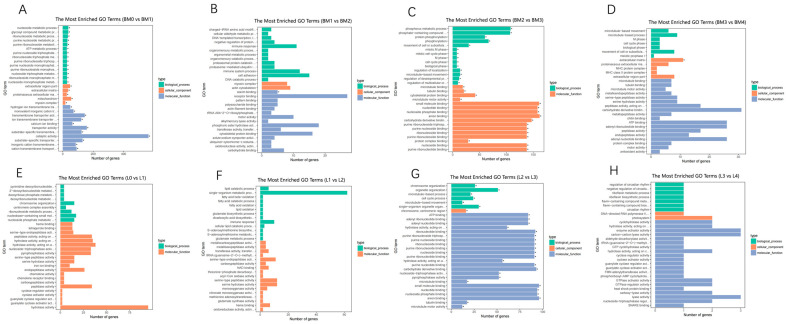
The top 30 GO terms of DEGs of each comparison library in the breast muscle and liver. Bars with asterisks were significant enrichments. 0, 1, 2, 3 and 4 weeks post-hatching. BM: breast muscle; L: liver. (**A**): BM0 vs. BM1; (**B**): BM1 vs. BM2; (**C**): BM2 vs. BM3; (**D**): BM3 vs. BM4; (**E**): L0 vs. L1; (**F**): L1 vs. L2; (**G**): L2 vs. L3; (**H**): L3 vs. L4.

**Figure 5 animals-12-01061-f005:**
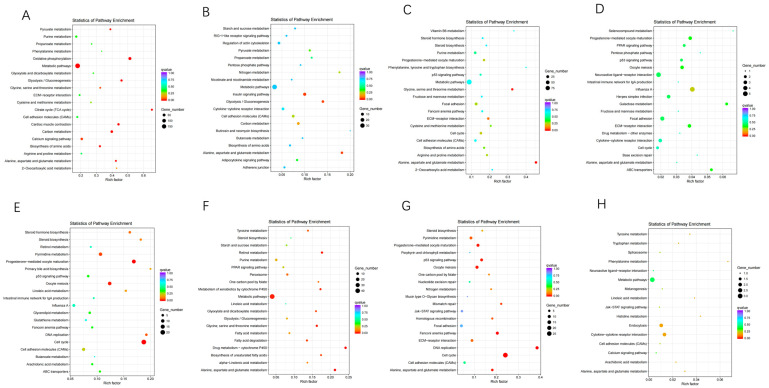
Top 20 functional enrichment pathways of each comparison library in breast muscle and liver. 0, 1, 2, 3 and 4 weeks post-hatching. BM: breast muscle; L: liver. (**A**): BM0 vs. BM1; (**B**): BM1 vs. BM2; (**C**): BM2 vs. BM3; (**D**): BM3 vs. BM4; (**E**): L0 vs. L1; (**F**): L1 vs. L2; (**G**): L2 vs. L3; (**H**): L3 vs. L4.

**Figure 6 animals-12-01061-f006:**
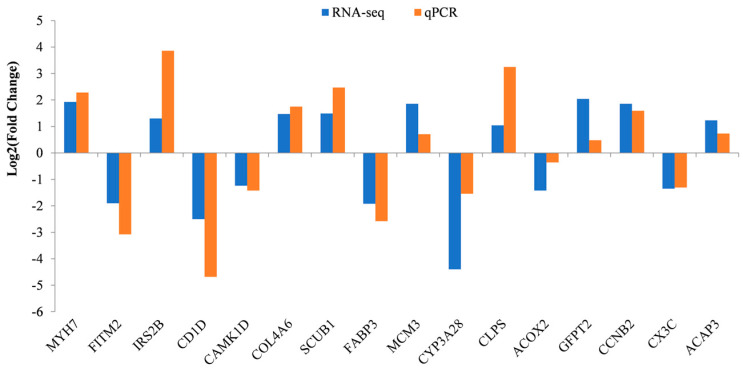
Expression of sixteen significant DEGs detected by RNA-Seq and validated by qPCR.

**Figure 7 animals-12-01061-f007:**
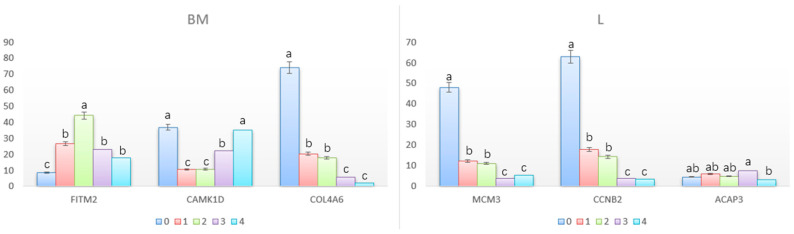
Temporal expression profile of 6 DEGs in breast muscle and liver. BM: breast muscle; L: liver. Bars with different lowercase letters were significantly different (*p* < 0.05).

**Table 1 animals-12-01061-t001:** Summary of sequencing reads aligned with the pigeon genome and annotated genes. 0, 1, 2, 3 and 4 weeks post-hatching. L: liver; BM: breast muscle.

Sample Name	L0	L1	L2	L3	L4
Raw reads	61,734,385	61,825,084	68,691,135	62,797,730	58,794,907
Clean reads	60,425,300	60,521,188	67,361,552	61,467,111	57,627,533
Total mapped	46,595,449 (77.11%)	46,975,350 (77.62%)	52,800,376 (78.38%)	47,8763,96 (77.89%)	45,009,613 (78.10%)
Multiple mapped	1,486,571 (2.46%)	1,128,089 (1.86%)	1,296,398 (1.92%)	1,110,525 (1.81%)	1,020,365 (1.77%)
Uniquely mapped	45,108,878 (74.65%)	45,847,261 (75.75%)	51,503,978 (76.46%)	46,765,871 (76.08%)	43,989,248 (76.33%)
**Sample Name**	**BM0**	**BM1**	**BM2**	**BM3**	**BM4**
Raw reads	59,696,172	60,023,444	60,920,173	58,235,371	60,401,530
Clean reads	58,190,291	58,469,628	59,430,898	56,592,143	58,867,155
Total mapped	43,975,943 (75.57%)	43,759,629 (74.84%)	45,075,823 (75.85%)	41,055,902 (72.55%)	43,344,616 (73.63%)
Multiple mapped	1,030,850 (1.77%)	1,273,665 (2.18%)	1,449,770 (2.44%)	1,433,186 (2.53%)	1,681,720 (2.86%)
Uniquely mapped	42,945,093 (73.80%)	42,485,964 (72.66%)	43,626,054 (73.41%)	39,622,716 (70.01%)	41,662,897 (70.77%)

## Data Availability

The data that supports the findings of this study are available from the corresponding author upon reasonable request.

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
