# Peer review of "Identification of Genes Related to Squab Muscle Growth and Lipid Metabolism from Transcriptome Profiles of Breast Muscle and Liver in Domestic Pigeon (Columba livia)"

_animals, 2022, doi:10.3390/ani12091061_

Round 1

Reviewer 1 Report

The research aimed to identify the genes related to squab muscle growth and lipid metabolism from transcriptome profiles of breast muscle and liver in domestic pigeon.

The introduction provide sufficient background information for readers not in the immediate field to understand the hypotheses. The reasons for performing the study are clearly defined. Also the study objectives are clearly defined. The experimental apparatus is appropriate to the aims of the study. Sufficient information is provided for a capable researcher to reproduce the experiment described. Appropriate references are cited where previously established methods are used. The results are clearly explained and presented in an appropriate format. The figures and tables show essential data and are easy to interpret. Appropriate statistical methods been used to test the significance of the results. The findings are properly described and discussed in the context of the published literature. The literature cited is balanced. The claims in the paper are sufficiently novel to warrant publication. The study represent a conceptual advance over previously published work.

Author Response

Dear reviewer,

  Thank you for reviewing our manuscript in your busy schedule, and we really appreciate for your compliments on our work. We will work harder on in-depth research.

Reviewer 2 Report

The manuscript by Yin and co-authors focusses on muscle growth and lipid metabolism in domestic pigeon during post-hatch development. The authors used RNAseq to look for differentially expressed genes in muscle and liver of developing pigeons. The main issue with the manuscript is n (biological replication) for RNAseq. Since there were samples collected from 5 different birds at each time point, I do not see any reason for sample pooling and running RNAseq as n=1.

There are few other issues with the manuscript that need to be addressed:

  1. What was the n for RNAseq? Where the sample collected pooled together? The methodology is not clear.
  2. Material and Methods: part 2.5: please provide references or websites for software used to analyze RNAseq data.
  3. Material and Methods, part 2.6: 2-delta delta Ct method is not a statistical method. Please correct and provide correct statistical method for real-time data analysis.
  4. First part of the results should be part of the data analysis in Material and Methods.
  5. All genes abbreviations should be spelled out when first used.
  6. 6, what was the n value for this data?
  7. Supplementary data were not provided with the manuscript.
  8. Validation of some of the genes found to be differentially expressed in muscle or liver by real-time expression profile during post-hatch development would be plus for the manuscript.

Author Response

Dear reviewer,

  we really appreciate for reviewing our manuscript in your busy schedule,  Thank you for your comments on our paper. After carefully studying your suggestions, we have revised the manuscript. we submit the revised manuscript in attachment. Hope these will more acceptable for publication.

Your sincerely!

Reviewer 3 Report

Dear Authors,

The reviewed article is very interesting for me (pigeons!), very well written, with smooth transitions between the topics discussed.

All comments are marked (directly) in the file. Please pay attention to unauthorized (wrong) citations and improve the citation format of the MS to the requirements (format) of Animals. I recommend accepting the manuscript for publication after applying the indicated corrections and taking a stance on the comments contained in the text.

Best regards,

Author Response

Dear reviewer,

  We really appreciate for reviewing our manuscript in your busy schedule. Thank you for your comments on our paper. After carefully studying your suggestions, we have revised the manuscript. we submit the revised manuscript in attachment. Hope these will make it more acceptable for publication.

Best regards!

Round 2

Reviewer 2 Report

I am accepting most of the responses to my questions and comments. however, still few things need to be clarified:

Point 1: based on your response there were 25 different libraries constructed and indexed for liver and muscle. So your n value for RNAseq was n=5. Or you have one library constructed and indexed from pooled samples for each time point?  Please verify and correct in manuscript. There is huge difference in results interpretation when n=1 or n=5.

Point 3. What  statistical method have you used for RT-PCR data analysis? Please add to the manuscript. None of your figures showing gene expression analysis have error bars or any indication of statistical analysis. Please provide accordingly.

Point 6. The question was about the numbers of replicates for each time point, in your case it should be n=5 (five birds per each time point).

Point 7. Please provide statistical analysis for your gene expression data.

Author Response

Dear reviewer,

  Thanks very much for taking your time to review this manuscript. I really appreciate all your comments and suggestions! Please find my  responses in attachment.
